# 3D-GPT: Procedural 3D Modeling with Large Language Models

## Abstract

In the pursuit of efficient automated content creation, procedural generation, leveraging modifiable parameters and rule-based systems, emerges as a promising approach. Nonetheless, it could be a demanding endeavor, given its intricate nature necessitating a deep understanding of rules, algorithms, and parameters. To reduce workload, we introduce 3D-GPT, a framework utilizing large language models (LLMs) for instruction-driven 3D modeling. 3D-GPT positions LLMs as proficient problem solvers, dissecting the procedural 3D modeling tasks into accessible segments and appointing the apt agent for each task. 3D-GPT integrates three core agents: the task dispatch agent, the conceptualization agent, and the modeling agent. They collaboratively achieve two objectives. First, it enhances concise initial scene descriptions, evolving them into detailed forms while dynamically adapting the text based on subsequent instructions. Second, it integrates procedural generation, extracting parameter values from enriched text to effortlessly interface with 3D software for asset creation. Our empirical investigations confirm that 3D-GPT not only interprets and executes instructions, delivering reliable results but also collaborates effectively with human designers. Furthermore, it seamlessly integrates with Blender, unlocking expanded manipulation possibilities. Our work highlights the potential of LLMs in 3D modeling, offering a basic framework for future advancements in scene generation and animation.

## 1 Introduction

In the metaverse era, 3D content creation serves as a catalyst for transformative progress, redefining multimedia experiences in domains like gaming, virtual reality, and cinema with intricately crafted models. Yet, designers often grapple with a time-intensive 3D modeling process, starting from basic shapes (*e.g.*, cubes, spheres, or cylinders) and employing software like Blender for meticulous shaping, detailing, and texturing. This demanding workflow concludes with rendering and post-processing to deliver the polished final model. While procedural generation holds promise with its efficiency in automating content creation through adjustable parameters and rule-based systems (Deitke et al., 2022; Greff et al., 2022; He et al., 2021; Jiang et al., 2018; Raistrick et al., 2023), it demands a comprehensive grasp of generation rules, algorithmic frameworks, and individual parameters. Furthermore, aligning these processes with the creative visions of clients, through effective communication, adds another layer of complexity. This underscores the importance of simplifying the traditional 3D modeling workflow to empower creators in the metaverse era.

LLMs have showcased exceptional language understanding capabilities, including planning and tool utilization (Imani et al., 2023; Zhang et al., 2023a; Gong et al., 2023; Zeng et al., 2022). Furthermore, LLMs demonstrate outstanding proficiency in characterizing object attributes, such as structure and texture (Menon & Vondrick, 2022; Pratt et al., 2022; Fan et al., 2023), enabling them to enhance details from rough descriptions. Additionally, they excel at parsing concise textual information and comprehending intricate code functions, while seamlessly facilitating efficient interactions with users. Driven by these extraordinary capabilities, we embark on exploring their innovative applications in procedural 3D modeling. Our primary objective is to harness the power of LLMs to exert control over 3D creation software in accordance with the requirements of clients.

In pursuit of this vision, we introduce 3D-GPT, a framework aimed at facilitating instruction-driven 3D content synthesis. 3D-GPT enables LLMs to function as problem-solving agents, breaking down

the 3D modeling task into smaller, manageable components, and determining when, where, and how to accomplish each segment. 3DGPT comprises three key agents: conceptualization agent, 3D modeling agent and task dispatch agent. The first two agents collaborate harmoniously to fulfill the roles of 3D conceptualization and 3D modeling by manipulating the 3D generation functions. Subsequently, the third agent manages the system by taking the initial text input, handling subsequence instructions, and facilitating effective cooperation between the two aforementioned agents.

By doing so, they work toward two key objectives. First, it enhances initial scene descriptions, guiding them towards more detailed and contextually relevant forms while adapting the textual input based on subsequent instructions. Second, instead of directly crafting every element of 3D content, we employ procedural generation, making use of adaptable parameters and rule-based systems to interface with 3D software. Our 3D-GPT is equipped with the capability to understand procedural generation functions and extract corresponding parameter values from the enriched text.

3D-GPT offers controllable and precise 3D generation guided by users' textual descriptions. It reduces the workload of manually defining each controllable parameter in procedural generation, particularly within complex scenes that encompass diverse aspects. Moreover, 3D-GPT enhances collaboration with users, making the creative process more efficient and user-centric. Furthermore, 3D-GPT seamlessly interfaces with Blender, granting users diverse manipulation capabilities: object transformations, material adjustments, primitive additions, object animations, mesh editing, and physical motion simulations. Based on our experiments, we posit that LLMs exhibit the potential to handle more intricate visual inputs. Our contributions are summarized as follows:

- Introducing 3D-GPT, a training-for-free framework designed for 3D scene generation. Our approach leverages the innate multimodal reasoning capabilities of LLMs, streamlining the efficiency of end-users engaged in procedural 3D modeling.
- Exploration of an alternative path in text-to-3D generation, wherein our 3D-GPT generates Python codes to control 3D software, potentially offering increased flexibility for real-world applications.
- Empirical experiments demonstrate the substantial potential of LLMs in terms of their reasoning, planning, and tool-using capabilities in 3D content generation.

## 2 RELATED WORK

### 2.1 TEXT-TO-3D GENERATION

With the recent advance in text-to-image generation modeling, there has been a growing interest in text-to-3D generation (Sanghi et al., 2022; Poole et al., 2022; Lin et al., 2023; Xu et al., 2023; Metzer et al., 2023; Wang et al., 2023; Xu et al., 2023; Mohammad Khalid et al., 2022; Jain et al., 2022). The common paradigm of them is to perform per-shape optimization with differentiable rendering and the guidance of the CLIP model (Radford et al., 2021) or 2D diffusion models (Rombach et al., 2022). For example, DreamFields (Jain et al., 2022) and CLIP-Mesh (Mohammad Khalid et al., 2022) explore zero-shot 3D content creation using only CLIP guidance. Dreamfusion (Poole et al., 2022) optimizes NeRF Mildenhall et al. (2021) with the guidance of a text-to-image diffusion model, achieving remarkable text-to-3D synthesis results. Further works in this direction have resulted in notable enhancements in visual quality (Lin et al., 2023; Melas-Kyriazi et al., 2023), subject-driven control (Raj et al., 2023; Metzer et al., 2023), and overall processing speed (Liu et al., 2023; Jain et al., 2022). Unlike the above approaches, our objective is not to generate conventional neural representations as the final 3D output. Instead, we utilize LLMs to generate Python code that controls Blender's 3D modeling based on the provided instructions.

### 2.2 LARGE LANGUAGE MODELS

Large language models (LLMs) are a promising approach to capture and represent the compressed knowledge and experiences of humans, projecting them into language space (Devlin et al., 2018; Raffel et al., 2020; OpenAI, 2023; Chowdhery et al., 2022; Bubeck et al., 2023). LLMs have consistently showcased remarkable performance extending beyond canonical language processing domains. They exhibit the capability to address intricate tasks that were once considered the exclusive domain of specialized algorithms or human experts. These tasks encompass areas such as mathematical reasoning (Imani et al., 2023; Wei et al., 2022), medicine (Jeblick et al., 2022; Yang et al.,

2023), and planning (Zhang et al., 2023a; Gong et al., 2023; Huang et al., 2023; 2022) Our work explores the innovative application of LLMs in 3D modeling, employing them to control 3D procedural generation.

# 3 3D-GPT

## 3.1 TASK FORMULATION

The overall objective is the generation of 3D content based on a sequence of relatively short natural language instructions, denoted as $\mathcal{L} = \langle L_i \rangle$. The initial instruction, designated as $L_0$, serves as a comprehensive description of the 3D scene, such as "*A misty spring morning, where dew-kissed flowers dot a lush meadow surrounded by budding trees*". Subsequent instructions are employed to modify the existing scene, as exemplified by instructions like "*transform the white flowers into yellow flower*" or "*translate the scene into a winter setting*" to add detail.

To accomplish this objective, we introduce a framework named 3D-GPT, which empowers LLMs to act as problem-solving agents. We point out that employing LLMs to directly create every element of 3D content poses significant challenges. LLMs lack specific pre-training data for proficient 3D modeling and, as a result, may struggle to accurately determine what elements to use and how to modify them based on given instructions. To address this challenge, we employ procedural generation to control the 3D content creation. This makes use of adaptable parameters and rule-based systems to interface with 3D software (*e.g.*, Blender) so as to efficiently conduct 3D modeling (Deitke et al., 2022; Greff et al., 2022; He et al., 2021; Jiang et al., 2018; Raistrick et al., 2023). Nevertheless, there are several challenges that remain such as identifying the correct procedures to call and mapping of language to API parameters. We solve these using multiple language agents as will be discussed below.

Our approach conceptualizes the 3D procedural generation engine as a set of functions, denoted as $\mathcal{F} = \{F_j\}$, where each function $F_j$ takes parameters $P_j$. For example, `add_trees(scene, density, distance_min, leaf_type, fruit_type)` will takes a built natural scene as input and adds base trees to it.

Within our 3D-GPT framework, for each language instruction $L_i$, we formulate the modeling task as first selecting the subset of relevant functions $\hat{\mathcal{F}} \subseteq \mathcal{F}$, and then inferring the corresponding parameters $P_j$ for each function $F_j$ in this subset. The ultimate aim is to ensure that the functions in $\hat{\mathcal{F}}$ collectively generates a 3D scene that aligns with the descriptions provided in $\mathcal{L}$. By adeptly addressing both function selection and parameter inference for every sub-instruction $L_i$, 3D-GPT generates a Python script file that allows Blender's 3D modeling environment to render high-quality scenes consistent with the instruction sequence $\mathcal{L}$.

## 3.2 MODELING TOOL PREPARATION

In our framework, we utilize Infinigen Raistrick et al. (2023), a Python-Blender-based procedural generator equipped with a rich library of generation functions. To empower LLMs with the ability to proficiently leverage Infinigen, we provide following crucial language prompts for each function $F_j$:

- **Documentation** ($D_j$): A comprehensive explanation of the function's purpose and clear description of it's parameters $P_j$ as one would find in standard API documentation.
- **API code** ($C_j$): Restructured and highly readable function code, ensuring that it is accessible and comprehensible for LLMs.
- **Auxiliary parameter information** ($I_j$): Outlines specific information required to infer the function parameters, thereby assisting LLMs in understanding the context and prerequisites of each function. For example, in the case of a flower generation function, $I_j$ indicates the required visual properties for rendering, such as flower color, flower petal appearance (e.g., size, curve, and length), and flower center appearance.
- **Usage examples** ($E_j$): Illustrative examples that demonstrate how to infer the parameter $P_j$ from the accompanying text descriptions and subsequently invoke the function. Continuing with the example of a flower generation function, $E_j$ includes a practical demonstration of how to infer the parameters and call the function based on input text like "a sunflower"

By providing LLMs with these resources, we enable them to leverage their generative competencies in planning, reasoning, and tool utilization. As a result, LLMs can effectively harness Infinigen for 3D generation based on language instructions in a seamless and efficient manner. In the context of our work, the function set $\mathcal{F}$ encompasses all functions and subfunctions within the Infinigen scene generation script, with the sole exception of the 'creatures' class. These functions play an indispensable role in our scene creation process. In the supplementary material, Section 6.4 presents a comprehensive list of all the functions by the script we utilized to construct the scenes. Additionally, we provide examples for using some of these functions in Section 6.8, Figure 14 and Figure 13.

## 3.3 Multi-agents for 3D Reasoning, Planing and Tool Using

With the necessary tool preparation (i.e., $D_j$, $C_j$, $I_j$ and $E_j$) in hand, 3D-GPT employs a multi-agent system to tackle the task of language-guided procedural 3D modeling. This system comprises three integral agents: (1) the task dispatch agent, (2) the conceptualization agent, and (3) the modeling agent, illustrated in Figure 1. Together, these agents decompose modeling task into manageable segments, with each agent specializing in distinct aspects: planning, 3D reasoning, and tool utilization. The task dispatch agent plays a pivotal role in the planning process. It leverages user instructions to query function documents and subsequently selects the requisite functions for execution. Once functions are selected, the conceptualization agent engages in reasoning to enrich the user-provided text description. Building upon this, the modeling agent deduces the parameters for each selected function and generates Python code scripts to invoke Blender's API, facilitating the creation of the corresponding 3D content. From there, images can be generated using Blender rendering capability.

**Task Dispatch Agent for Planing.** The task dispatch agent, armed with comprehensive information of all available functions $\mathcal{F}$ as described above, efficiently identifies the requisite functions for each instructional input. For example, when presented with an instruction such as "*translate the scene into a winter setting*", it pinpoints functions like `add_snow_layer` and `update_trees`. This pivotal role played by the task dispatch agent is instrumental in facilitating efficient task coordination between the conceptualization and modeling agents. Without it, the conceptualization and the modeling agents have to analyze all provided functions $\mathcal{F}$ for each given instruction. This not only increases the workload for these agents but also extends processing time and can potentially lead to undesired modifications. The communication flow between the LLM system, the user, and the task dispatch agent is outlined as follows:

> — **LLM System**: *You are a proficient planner for selecting suitable functions based on user instructions. You are provided with the following functions: $\langle (F_j^{name}, F_j^{usage}) \rangle$. Below are a few examples of how to choose functions based on user instructions: $\langle E_j^{task\_dispatch} \rangle$.*
> — **User**: *My instruction is: $\langle L_i \rangle$.*
> — **Task Dispatch Agent**: *Given the instruction $\langle L_i \rangle$, we determine the sublist of functions $\hat{\mathcal{F}}$ that need to be used for 3D modeling.*

Here $\langle (F_j^{name}, F_j^{usage}) \rangle$ represents a list of function names and concise function usage descriptions for all available functions and examples $\langle E^{task\_dispatch} \rangle$ provide guided examples for prompt-based instructions. A example is provided in the supplementary Figure 10.

**Conceptualization Agent for Reasoning.** The user instruction may not explicitly provide sufficient details needed for modeling. For instance, consider the instruction, "*a misty spring morning, where dew-kissed flowers dot a lush meadow surrounded by budding trees*". Here many necessary details required function parameters such as tree branch length, tree size, and leaf type, are not directly stated in the given text. When instructing the modeling agent to infer parameters directly, we observed that it tends to provide simplistic solutions, such as using default or copying values from prompting examples. This reduces diversity in generation and complicates parameter inference.

To address this issue, we introduce the conceptualization agent which collaborates with the task dispatch agent to augment the user-provided text description ($L_i$). After the task dispatch agent selects the required functions, we send the user input text and the corresponding function-specific information to the conceptualization agent and request augmented text. For each function $F_j$, it enriches $L_i$ into detailed appearance descriptions $L_i^j$. The communication between the system and the Conceptualization Agent for instruction $\langle L_i \rangle$ and function $\langle F_j \rangle$ is as follows:

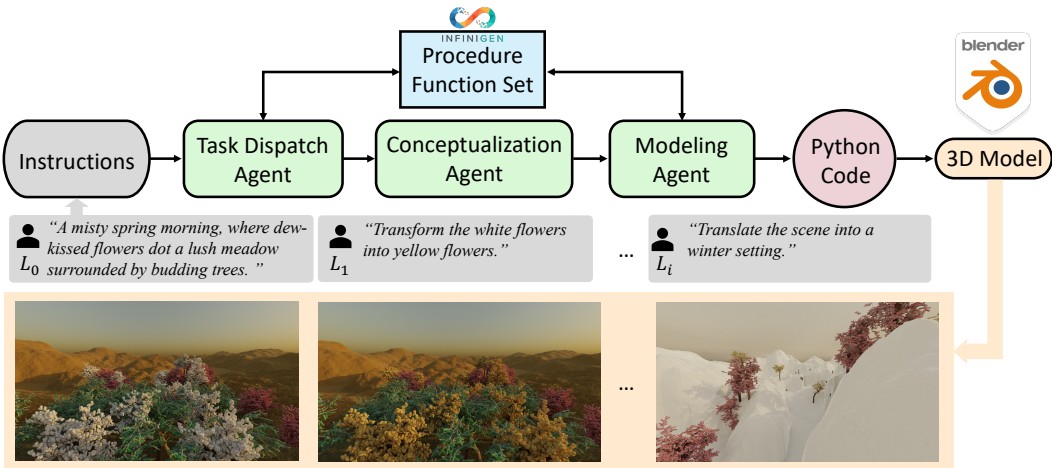

Figure 1: **3D-GPT Overview.** 3D-GPT employs LLMs as a multi-agent system with three collaborative agents for procedural 3D generation. These agents consult documents from the procedural generator, infer function parameters, and produce Python code. The generated code script interfaces with Blender's API for 3D content creation and rendering.

---

**— LLM System**: *You are a skilled writer, especially when it comes to describing the appearance of objects and large scenes. Given a description $\langle L_i \rangle$, provide detailed descriptions for the following information $\langle I_j \rangle$. For terms not mentioned in the description, use your imagination to ensure they fit the text description.*
**— Conceptualization Agent**: *Given the $\langle L_i \rangle$ and requested information $\langle I_j \rangle$, the extended description is: $\langle \widehat{L_i^j} \rangle$.*

---

We have illustrated a communication example in Figure 11 within the supplementary material.

**Modeling Agent for Tool Using.** After conceptualization, the 3D modeling processing is invoked to convert the detailed human language to machine-understandable language. In our framework, our modeling agent employs the functions of procedural modeling in the library to create a realistic 3D model. For each function $F_j$ and user instruction $L_i$, the task dispatch agent receive augmented context $\widehat{L_i^j}$ from the conceptualization agent. For each function $F_j$, we have the code $C_j$, function documentation $D_j$, and one usage example $E_j$. The modeling agent utilizes this information to select the appropriate functions and deduce the corresponding parameters. Subsequently, the modeling agent generates Python code that calls the selected function in the right context (e.g., within a loop), passing in parameters inferred from the text and of the appropriate data type.

The communication between System and Modeling Agent are based on the following pattern:

---

**— LLM System**: *You are a good 3D designer who can convert long text descriptions into parameters, and is good at understanding Python functions to manipulate 3D content. Given the text description $\langle \widehat{L_i^f} \rangle$, we have the following function codes $\langle C_j \rangle$ and the document for function $\langle D_j \rangle$. Below is an example bout how to make function calls to model the scene to fit the description: $\langle E_j^{modeling} \rangle$. Understand the function, and model the 3D scene that fits the text description by making a function call.* **— Modeling Agent**: *Given the description $\langle \widehat{L_i^j} \rangle$, we use the following functions: ..., and their respective parameter values ... are adopted.*

---

We have illustrated a communication example in Figure 12 within the supplementary material.

**Blender Rendering.** The modeling agent ultimately constructs the Python function calls with inferred parameters, which are supplied to Blender for controlling view ports and rendering, and thereby resulting in production of the final 3D mesh and RGB results.

**Implementation Detail.** Our implementation relies on the Infinigen Raistrick et al. (2023) API, and the specific function set $\mathcal{F}$ utilized in our work is available in the generation script provided in

*"A vibrant autumn forest, with trees ablaze in shades of red, orange, and gold, as a gentle breeze rustles the fallen leaves."*

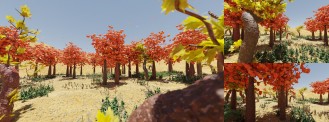

*"A misty spring morning, where dew-kissed flowers dot a lush meadow surrounded by budding trees."*

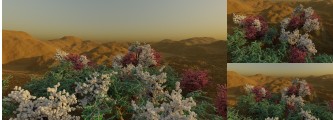

*"A serene winter landscape, with snow-covered evergreen trees and a frozen lake reflecting the pale sunlight."*

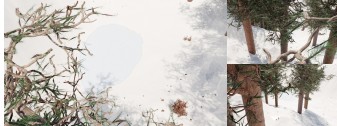

*"The mountains, majestic and snow-capped, stood like sentinels guarding the vast expanse of the valley, their peaks disappearing into the swirling mist that clung to their rugged slopes."*

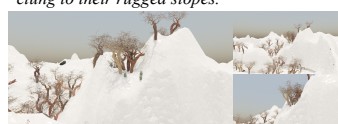

*"The desert, an endless sea of shifting sands, stretched to the horizon, its rippling dunes catching the golden rays of the setting sun, creating an ever-changing landscape of shadows and light."*

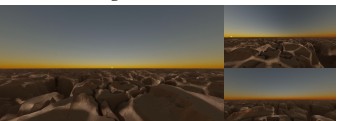

*"The lake, serene and glassy, mirrored the cloudless sky above, reflecting the surrounding mountains and the graceful flight of a heron, as lily pads floated like emerald jewels upon its tranquil surface."*

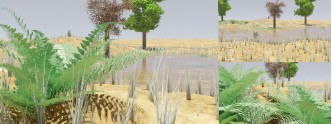

Figure 2: **Visual Examples of Instruction-Based 3D Scene Generation.** 3D-GPT can construct large 3D scenes that align with the provided initial instruction. We demonstrate that the rendered images contain various visual factors in line with the given instructions.

Supplementary Material, as detailed in Section 6.4. We have developed our system using the OpenAI API, and the code implementation for our modeling agent can also be found in Supplementary Material, Section 6.4. This code demonstrates the ease with which our system can be implemented.

## 4 EXPERIMENTS

Our experimentation begins by showcasing the proficiency of 3D-GPT in consistently generating results that align with user instructions, encompassing scenarios involving both large scenes and individual objects. Subsequently, we delve into specific examples to illustrate how our agents effectively comprehend tool functionalities, access necessary knowledge, and employ it for precise control. To deepen our understanding, we conduct an ablation study to systematically examine the contributions of each agent within our multi-agent system.

### 4.1 3D MODELING

**Large Scene Generation.** We investigate the capability of 3D-GPT to control modeling tools based on scene descriptions *without any training*. To conduct this experiment, we generated 100 scene descriptions using ChatGPT with the following prompt: "*You are a good writer, provide* 10 *different natural scene descriptions for me*". We collected responses to this prompt 10 times to form our dataset. In Figure 2, we present the multi-view rendering results of 3D-GPT. These results indicate that our approach is capable of generating large 3D scenes that generally align well with the provided text descriptions, showcasing a noticeable degree of diversity. Notably, all 3D outcomes are directly rendered using Blender, ensuring that all meshes are authentic, thereby enabling our method to achieve absolute 3D consistency and produce real ray-tracing rendering results.

**Fine-detail Control for Single Class.** Apart from generating large scenes from concise descriptions, we assess the capabilities of 3D-GPT for modeling objects. We evaluate crucial factors such as curve modeling, shape control, and an in-depth understanding of object appearances. To this end, we report the results of fine-grained object control. This includes nuanced aspects such as object curves, key appearance features, and color, all derived from input text descriptions. We employ random prompts to instruct GPT for various real-world flower types. As depicted in Figure 3, our method adeptly models each flower type, faithfully capturing their distinct appearances. This study underscores the potential of 3D-GPT in achieving precise object modeling and fine-grained attribute control of object types and visual characteristics.

**Subsequence Instruction Editing.** Here, we test the ability of 3D-GPT for effective human-agent communication and task manipulation. In Figure 4, we observe that our method can comprehend

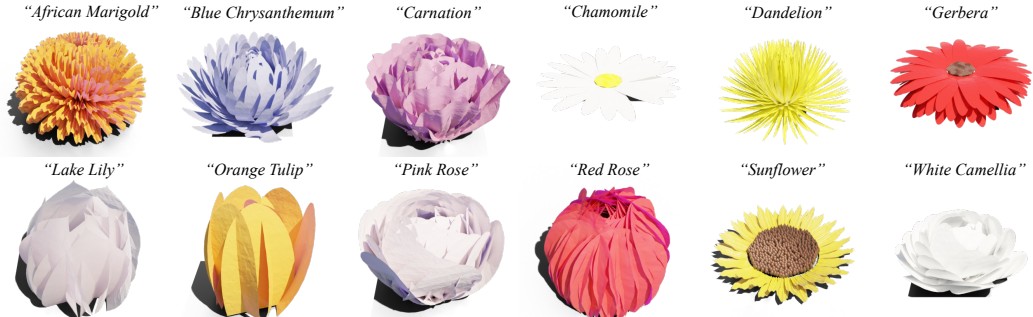

Figure 3: **Single Class Control Result.** Our method effectively acquires the necessary knowledge for modeling, enabling precise object control in terms of shape, curve, and key appearance capture. The generated results closely align with the given text.

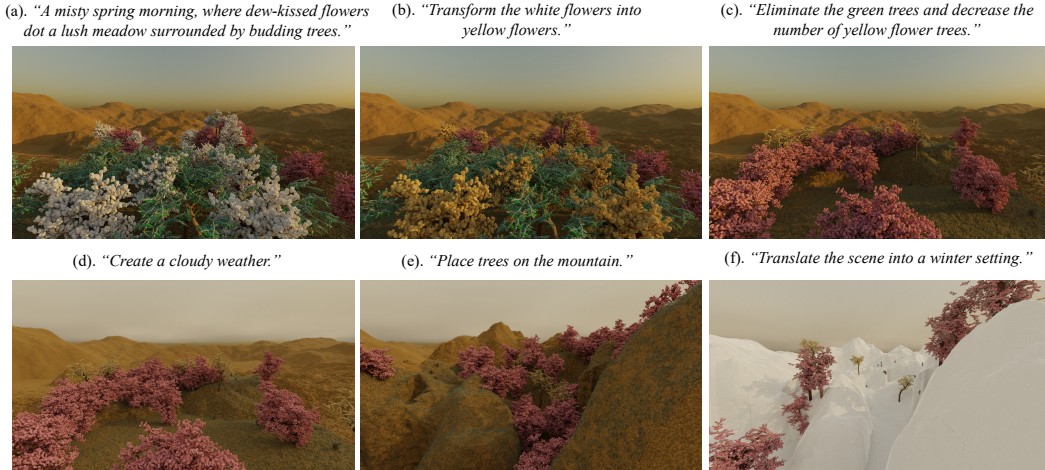

Figure 4: **Subsequence Instruction Editing Result.** (a) Initial instruction-generated scene. (b)-(f) Sequential editing steps with corresponding instructions. Our method enables controllable editing and effective user-agent communication.

subsequence instructions and make accurate decisions for scene modification. Note that, unlike the existing text-to-3D methods, 3D-GPT maintains a memory of all prior modifications, thereby facilitating the connection of new instructions with the scene's context. Furthermore, our method eliminates the need for additional networks for controllable editings Zhang et al. (2023b). This study underscores the efficiency and versatility of 3D-GPT in adeptly handling complex subsequence instructions for 3D modeling.

**Individual Function Control**  To evaluate the effectiveness of 3D-GPT in tool utilization, we present an illustrative example that highlights our method's ability to control individual functions and infer parameters. Figure 5 exemplifies the capability of 3D-GPT to model sky appearances based on input text descriptions. It is worth noting that the function responsible for generating the sky texture does not directly correlate color information with sky appearance. Instead, it relies on the Nishita-sky modeling method, which requires a profound understanding of real-world sky and weather conditions, considering input parameters. Our method adeptly extracts crucial information from the textual input and comprehends how each parameter influences the resulting sky appearance, as evident in Figure 5 (c) and (d). These results demonstrate that our method can effectively use individual functions as well as infer corresponding parameters.

(a). *"The sun shines brightly in a clear blue sky. Not a cloud in sight."*

(b). *"Thick gray clouds cover the entire sky, diffusing the sunlight."*

(c). *"The sky appears hazy, with a soft, diffuse sunlight filtering through."*

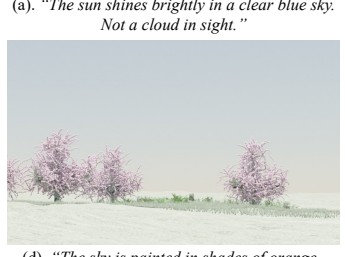
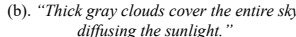
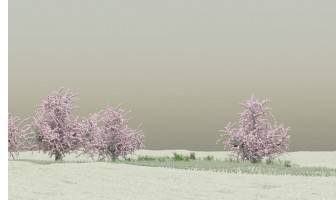
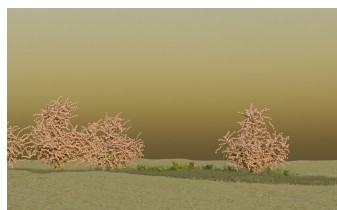

(d). *"The sky is painted in shades of orange and pink as the sun sets behind the island."*

(e). *"The sky is a mixture of soft purples and pinks as the sun rises."*

(f). *"A canopy of stars fills the dark night sky, with the island bathed in moonlight."*

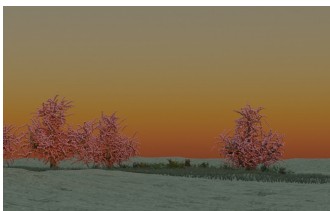
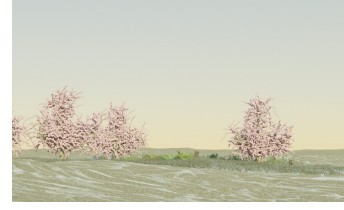
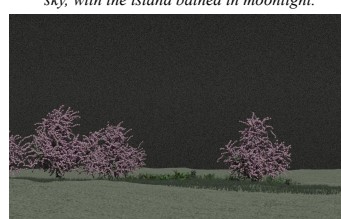

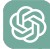

(a). Solution: From the description of "clear blue sky" and "not a cloud in sight", we can infer that the sun_intensity should be high, the sun_elevation should be around 90 (daytime), the sun_rotation can be set to 0. The air_density and dust_density should be low, so we can set them to 1 and 0 respectively. Since there are no clouds, the cloud_density should be 0. To make the sky appear bluer, we can increase the ozone value, so let's set it to 2. We can model the sky by calling the following function:

```python
sky_texture_node(sun_intensity='high', sun_elevation=90, sun_rotation=0, air_density=1, dust_density=0, ozone=2, cloud_density=0)
```

Figure 5: **Single Function Control Result.** Visual result (top) and modeling agent response example (bottom). Our method demonstrates a high degree of accuracy in inferring algorithm parameters, even when they do not possess a direct connection to visual appearance.

## 4.2 ABLATION STUDY

We conduct separate ablation studies for the Conceptualization Agent and Task Dispatch Agent. Our assessment focused on CLIP scores (Radford et al., 2021), failure rates, and parameter diversity, quantified using the categorical Shannon Diversity Index. The CLIP score measures the alignment between text and generated images. The failure rate represents the percentage of system failures due to issues such as incorrect datatypes, wrong response patterns, or missing parameters from the Modeling Agent. Parameter diversity aims to gauge the diversity of generated outputs.

| Metrics/ Method | CLIP Score |
| --- | --- |
| w/o TDA | 22.79 |
| Ours | 29.16 |

(a) **Ablation Study of Task Dispatch Agent.**

| Metrics/ Method | CLIP Score | Failure Rate | Parameter Diversity |
| --- | --- | --- | --- |
| w/o CA | 21.51 | 3.6% | 6.32 |
| Ours | 30.30 | 0.8% | 7.34 |

(b) **Ablation Study of Conceptualization Agent.**

Table 1: **Ablation Study.** "w/o CA" indicates without the Conceptualization Agent, "w/o TDA" indicates without the Task Dispatch Agent.

**Case Study of Task Dispatch Agent.** For the Task Dispatch Agent, the CLIP score is measured using 100 initial scene descriptions, each appended with one additional subsequence instruction for each scene. Table 1 (a) shows that without the Task Dispatch Agent, the CLIP score dropped from 29.16 to 22.79. It is important to note that the Task Dispatch Agent primarily impacts the performance of subsequence instructions, as all functions are utilized for the initial instruction. These findings underscore the pivotal role of the Task Dispatch Agent in managing communication flow.

**Case Study of conceptualization Agent.** For the Conceptualization Agent, the CLIP score is measured using 100 initial scene descriptions. Table 1 (b) displays the results, indicating that without the

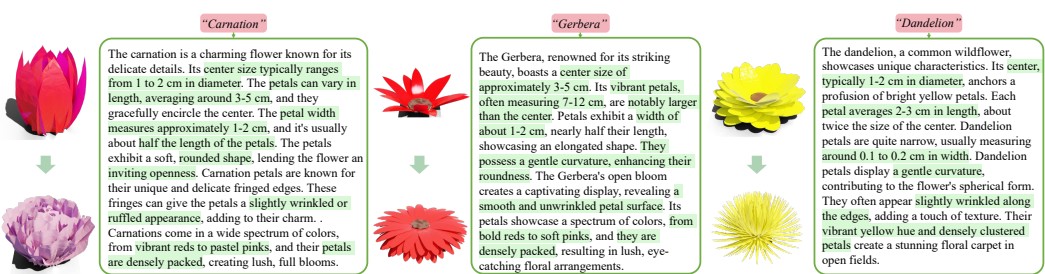

Figure 6: **Conceptualization Agent Case Study.** The enriched textual evidence demonstrates that the Conceptualization Agent provides essential knowledge for parameter inference (highlighted in green). For each subfigure, we compare the 3D model without (Top) and with (Bottom) agent. The models generated with the agent better match the text description than those without it .

Conceptualization Agent, both text alignments (CLIP score) and parameter diversity decreased significantly. Moreover, the failure rate increased substantially, which adversely impacts the efficiency of the entire modeling process. Figure 6 illustrates how the Conceptualization Agent facilitates the acquisition of essential knowledge for 3D modeling, providing a visual comparison of results with and without its involvement. When the Conceptualization Agent is engaged, the generated results closely align with the appearance of the intended flower type, highlighting its invaluable contribution to elevating overall 3D generation quality and fidelity.

## 5 DISCUSSION AND CONCLUSION

We have introduced 3D-GPT, a novel training-free framework for instruction-driven 3D modeling seamlessly integrated with procedural generation. Leveraging the capabilities of LLMs, 3DGPT aims to enhance human-AI communication in the context of 3D design. Our approach involves the collaborative efforts of three agents functioning as a cohesive 3D modeling team, ultimately yielding a 3D modeling file as output, as opposed to conventional 3D neural representations. Moreover, our method consistently delivers high-quality results, showcases adaptability to expansive scenes, ensures 3D consistency, provides material modeling and editing capabilities, and facilitates real ray tracing for achieving lifelike visualizations. Our empirical experiments show the potential of LLMs for reasoning, planning, and tool using in procedural 3D modeling.

**Limitations and Potential Directions.** While our framework has demonstrated promising 3D modeling results closely aligned with user instructions, it is essential to acknowledge several limitations: 1) Limited curve control and shading design: Currently, our framework lacks advanced capabilities for precise curve control and intricate shading design. Tasks involving the manipulation of tree branches or the blending of colors for leaf textures remain challenging. 2) Dependence on procedural generation algorithms: the effectiveness of our framework is contingent on the quality and availability of procedural generation algorithms. This reliance may limit results in specific categories, such as hair and fur. 3) Handling multi-modal instructions: challenges arise in processing multi-modal instructions, including audio and image inputs, potentially leading to information loss. These limitations offer valuable insights for shaping future research and development in the field. We highlight three compelling directions for future investigation:
*LLM 3D Fine-Tuning:* It is promising to fine-tune LLMs to enhance their capabilities in geometry control, shading design, and fine-texture modeling. This refinement will make LLMs more adept at handling intricate 3D modeling tasks and grant greater creative control over the resulting 3D scenes.
*Autonomous Rule Discovery:* Building on the demonstrated tool-making capabilities of LLMs, one direction is to develop an autonomous 3D modeling system that reduces human involvement. This could empower LLMs to autonomously discover generation rules for new object classes and scenes, thus expanding the creative potential.
*Multi-Modal Instruction Processing:* To achieve more comprehensive and expressive 3D modeling based on varied user inputs, it is crucial to enhance the system's ability to comprehend and respond to multi-modal instructions. This would facilitate richer and more diverse 3D modeling outcomes, shaped by a broader spectrum of user inputs.

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

## 6 APPENDIX

### 6.1 ADDITIONAL RESULT

We kindly request the reader to consider visiting `https://anonymous0888.github.io/3DGPT/3dgpt.html` to view our high-quality 3D results.

### 6.2 ADDITIONAL ABLATION STUDY

We conduct three distinct ablation studies examining prompting components (Table 2), various Large Language Models (Table 3), and example numbers (Table 4).

Table 2: Ablation Study of Prompting Components D/C/I/E.

| Metrics/Method | CLIP Score | Failure Rate | Parameter Diversity |
|---|---|---|---|
| w/o D | 20.7 | 4.2% | 6.94 |
| w/o C | 28.4 | 1.8% | 6.74 |
| w/o I | 21.6 | 1.4% | 6.38 |
| w/o E | 24.5 | 3.4% | 7.89 |
| Ours | 30.3 | 0.8% | 7.34 |

Table 3: Ablation Study of Different Large Language Model.

| Metrics/Model | CLIP Score | Failure Rate | Parameter Diversity |
|---|---|---|---|
| LLAMA2 | 29.7 | 1.4% | 6.97 |
| GPT4 | 31.2 | 0.6% | 7.23 |
| GPT3.5 | 30.3 | 0.8% | 7.34 |

Table 4: Ablation Study of Example Number.

| Metrics/Shots | CLIP Score | Failure Rate | Parameter Diversity |
|---|---|---|---|
| 0 | 24.5 | 3.4% | 7.89 |
| 1 | 30.3 | 0.8% | 7.34 |
| 2 | 30.1 | 1.0% | 7.23 |
| 3 | 30.2 | 0.8% | 6.93 |

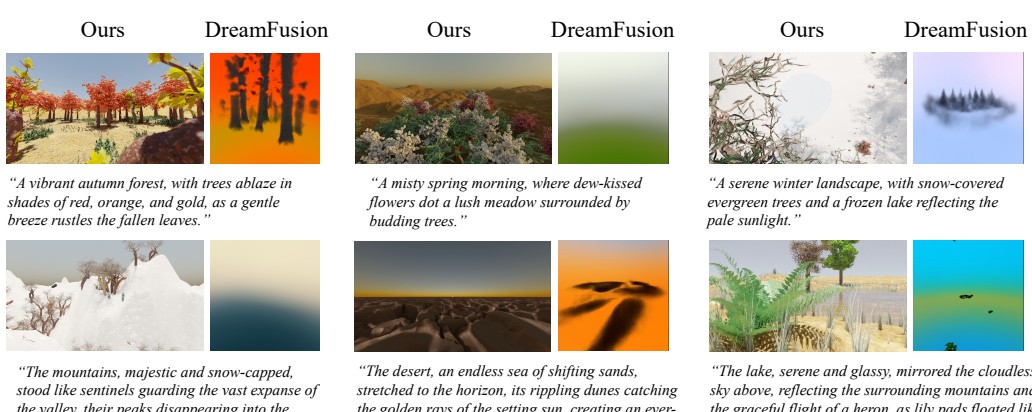

| Ours | DreamFusion | Ours | DreamFusion | Ours | DreamFusion |

*"A vibrant autumn forest, with trees ablaze in shades of red, orange, and gold, as a gentle breeze rustles the fallen leaves."*

*"A misty spring morning, where dew-kissed flowers dot a lush meadow surrounded by budding trees."*

*"A serene winter landscape, with snow-covered evergreen trees and a frozen lake reflecting the pale sunlight."*

*"The mountains, majestic and snow-capped, stood like sentinels guarding the vast expanse of the valley, their peaks disappearing into the swirling mist that clung to their rugged slopes."*

*"The desert, an endless sea of shifting sands, stretched to the horizon, its rippling dunes catching the golden rays of the setting sun, creating an ever-changing landscape of shadows and light."*

*"The lake, serene and glassy, mirrored the cloudless sky above, reflecting the surrounding mountains and the graceful flight of a heron, as lily pads floated like emerald jewels upon its tranquil surface."*

Figure 7: **Comparison with Dreamfusion (Scene).**

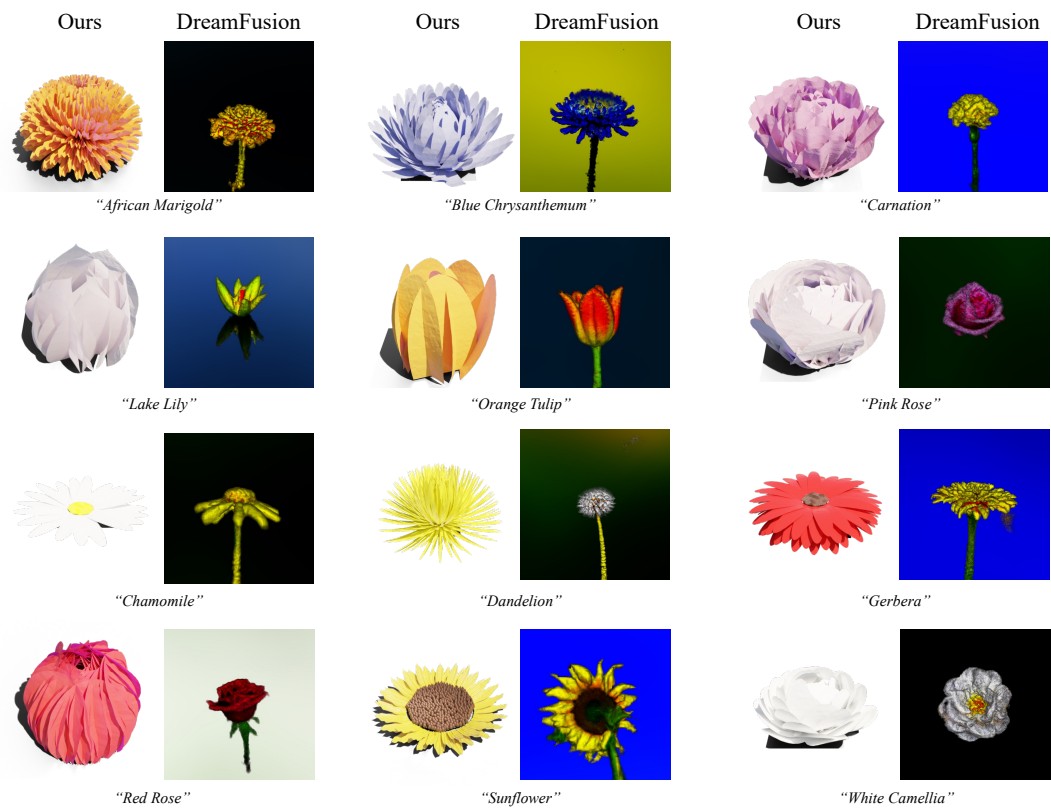

Figure 8: **Comparison with Dreamfusion (Single Object).**

## 6.3 COMPARISON WITH TEXT-TO-3D

We offer side-by-side comparisons with a state-of-the-art Text-to-3D method Dreamfusion (Poole et al., 2022) for single objects and scene, as illustrated in Figure 8 and Figure 7, respectively.

## 6.4 IMPLEMENTATION DETAIL

We provide the code that enumerates the functions within Infinigen used in constructing our 3D scene. While a similar Python file can be found at `https://github.com/princeton-vl/infinigen/blob/main/worldgen/generate.py`, our script is uniquely tailored to work with a control dictionary generated by three agents, enabling controllable scene generation.

```
def compose_scene(terrain, control_dict={}):
    """
    Give a base terrain and control dictionary, build the 3D scene.
    ----------
    terrain: Empty mesh
    control_dict: control dictionary generated by the agents that use to
    control each function parameters.
    Returns
    -------
    """
    add_coarse_terrain(terrain,control_dict)
    add_trees(control_dict)
    add_bushes(control_dict)
    add_clouds(control_dict)
    add_boulders(control_dict)
    add_glowing_rocks(control_dict)
    add_kelp(control_dict)
```

```
17    add_cactus(control_dict)
18    add_rocks(control_dict)
19    add_ground_leaves(control_dict)
20    add_ground_twigs(control_dict)
21    add_chopped_trees(control_dict)
22    add_grass(control_dict)
23    add_monocots(control_dict)
24    add_ferns(control_dict)
25    add_flowers(control_dict)
26    add_corals(control_dict)
27    add_leaf_particles(control_dict)
28    add_rain_particles(control_dict)
29    add_dust_particles(control_dict)
30    add_marine_snow_particles(control_dict)
31    add_snow_particles(control_dict)
32
33    return
```

Listing 1: Agent Implementation Example

We offer the code to implement the modeling agent as a demonstration of how to utilize the OpenAI API for implementing our agent.

```
1  def modeling_function_call(text_description,function_description,
       function,
2                            function_document,example,max_tokens=2000,
       temperature=0.3,history=[]):
3        """
4        Give a short text, call the given functions to generation objects/
       scene to fit the given text description
5        Parameters
6        ----------
7        text_description: short user given text.
8        function_description: short function description.
9        function: python code.
10       function_document: the detail description of the function.
11       example: example of how to use the function.
12       max_tokens: max tokens for the detailed text.
13       temperature
14       Returns
15       -------
16       response from the agent that contains the function calls.
17       """
18       if(len(history)==0):
19          history = [
20             {"role":"system", "content":"You are a good 3D designer who can
        convert long text descriptions into parameters, and is good at
       understanding Python functions to manipulate 3D content. "},
21          ]
22       messages = history
23
24       text = f"""We have the following function codes  {
       function_description} to control blender by python : {function}.
25          Following are the document for function: {function_document}.
26          Below is an example bout how to make function calls to model the
       scene to fit the description: {example}.
27          Question: Given the text description:    {text_description}
       analysis the function parameter and call the function to {
       function_description}"""
28       messages.append({"role": "user","content":text})
29
30       conceptualization_augmentation_model = openai.ChatCompletion.create
       (
31          model = "gpt-3.5-turbo",
32          temperature = temperature,
```

```
33        max_tokens = max_tokens,
34        messages = messages
35    )
36
37    conceptualization_text = conceptualization_augmentation_model.
    choices[0].message["content"]
38    return conceptualization_text,messages
```

Listing 2: Agent Implementation Example

### 6.5 DISCUSSION OF FUNCTION SET SIZE

Assessing the impact of function set size and parameter count can be complex due to variations in function significance. In Figure 9, we can see that removing rock modeling has a minimal impact on the modeling outcome. On the other hand, eliminating the water modeling function makes river modeling impossible, leading to a significant reduction in the alignment between the generated 3D scene and the text description.

(a). Remove rock modeling    (b). Remove water modeling    (c). Full function set

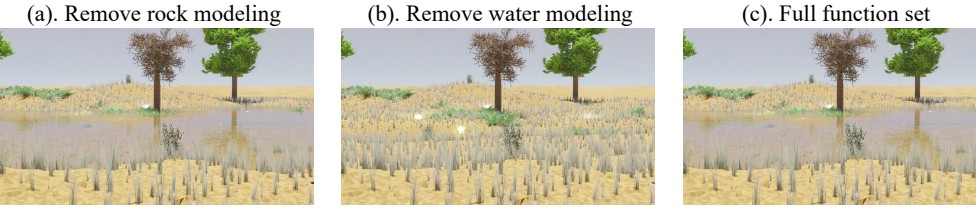

*"The lake, serene and glassy, mirrored the cloudless sky above, reflecting the surrounding mountains and the graceful flight of a heron, as lily pads floated like emerald jewels upon its tranquil surface."*

Figure 9: **Ablation study of Different Function Set**

### 6.6 ABLATION STUDY DETAILS

We conduct separate ablation studies for the Conceptualization Agent and Task Dispatch Agent, evaluating their performance based on CLIP scores, failure rates, and parameter diversity.

We use the implementation of Radford et al. (2021) to calculate CLIP score. It measures cosine similarity in the CLIP hidden space, indicating the alignment between generated images and text descriptions. For 3D scene CLIP scores, we placed a camera at the scene's center, capturing 50 images by rotating the camera 360 degrees. In the Conceptualization Agent evaluation, we used only the initial instruction. For the Task Dispatch Agent, both the initial instruction and one subsequence instruction were used.

The failure rate reflects the Modeling Agent's response. Failures can occur when the method cannot extract the correct pattern via the parser, generates data with an incorrect datatype, or omits/adds parameters to the function call. We evaluated five functions for each scene, recording the failure rate. We also assessed 500 parameters per scene, employing the Shannon Diversity Index to measure parameter diversity, and categorizing float parameters into 100 bins.

### 6.7 AGENT RESPONSES

Figure 10, Figure 11 and Figure 12 demonstrate how the system communicates with Task Dispatch Agent, Conceptualization Agent, and Modeling Agent, respectively. The Task Dispatch Agent example is deliberately simplified, featuring only four functions to ensure conciseness. In contrast, the examples for the Conceptualization Agent and Modeling Agent are derived from our comprehensive implementation.

You are a proficient planner for selecting suitable functions based on user instructions. You are provided with the following functions:
1.  generate_terrain(): model the 3D terrain.
2.  generate_tree(): generate trees with parameters that controls the tree appearance and place them on the terrain surface.
3.  add_grass(): add grass to the terrain surface()
4.  apply_snow_layer(): add snow to the surface of all objects.
Below are examples of how to choose functions based on user instructions:
User: My instruction is: "Change weather to snowy." Answer: Based on the function provided above and the user instruction, the apply_snow_layer() function from above function set is suitable to call. It will add snow to all objects covered by snow and make it look like a snowy scene. Other functions (1,2,3) are not relevant to the user instruction. Given the instructions, we determine the sublist of functions that need to be used for 3D modeling: [4]
User: My instruction is: "Change tree leaf to yellow."

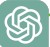

Based on the user instruction, "Change tree leaf to yellow," you can use the generate_tree() function to generate trees with yellow leaves. You can modify the parameters of the generate_tree() function to control the appearance of the trees, including the color of their leaves. Other functions (1, 3, 4) are not relevant to changing the color of tree leaves. Given the instructions, we determine the sublist of functions that need to be used for 3D modeling:
[2]
So, the suitable function to call in this case is generate_tree() with the appropriate parameters for yellow leaves.

Figure 10: **Communication Between System and Task Dispatch Agent.**

Given a text description "A peaceful clearing in a dense forest, blanketed in fresh, untouched snow. Tall birch trees, their white bark standing out starkly against the dark evergreens and the gray, overcast sky, form a natural cathedral.", provide detailed descriptions for the following information: "possible daytime, sun intensity, air condition, sky appearance, cloud density.".
For term not mentioned in the description, use your imagination to ensure they fit the text description.

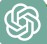

In this peaceful clearing, the scene is bathed in the soft glow of early morning light. The sun, just beginning to rise above the horizon, casts a gentle, golden hue across the landscape. Its intensity is subdued, as if still gathering strength to break through the thick canopy of trees that surround the clearing.
The air in the clearing is crisp and cool, carrying with it a hint of winter's bite. It is still and calm, devoid of any breeze or gusts that might disturb the tranquility of the scene. The air feels pure and invigorating, as if it has been untouched by human presence.
Looking up towards the sky, the overcast conditions create a muted, gray backdrop. The clouds hang low, their edges blending seamlessly with the horizon, creating a sense of enclosure within the clearing. The sky appears vast, yet somehow intimate, as if it is a part of the natural cathedral formed by the towering birch trees.
The cloud density is moderate, with the clouds spread out across the sky in a gentle, billowing fashion. They are not heavy or oppressive, but rather wispy and ethereal, adding a touch of mystery to the scene. As the day progresses, the clouds shift and change shape, creating a dynamic and ever-evolving canvas above the clearing.
Overall, the daytime in this clearing is characterized by a serene ambiance, with the sun's gentle rays filtering through the dense forest and the sky adorned with a delicate veil of clouds. It is a scene that invites contemplation and reflection, a sanctuary of tranquility amidst the wild beauty of nature.

Figure 11: **Communication Between System and Conceptualization Agent.**

## 6.8   VISUAL RESULT

Figure 13 and Figure 14 show the prompts for modelling trees and sky. The visual control result for a full scene is illustrated in Figure 15.

We have the following function codes modelling the sky appearance using nishita method to control blender by python :
```python
import nishita_sky_modelling
  def sky_texture_node(sun_intensity,sun_elevation,sun_rotation,air_density,dust_density, ozone,cloud_density):
      texture = nishita_sky_modelling(sun_intensity,sun_elevation,sun_rotation,air_density,dust_density, ozone,cloud_density)
      return
```
Following are the document for function: "
input:
 sun_intensity: Multiplier for sun disc lighting. (choose from 'low','median','high')
 sun_elevation: Rotation of the sun from the horizon (in degrees). (0:sunset,sun rising, 90:daytime,-10:night)
 sun_rotation: Rotation of the sun around the zenith (in degrees).
 air_density: density of air molecules. (0 no air, 1 clear day atmosphere, 2 highly polluted day)
 dust_density: density of dust and water droplets. (0 no dust, 1 clear day atmosphere, 5 city like atmosphere, 10 hazy day)
 ozone: density of ozone molecules; useful to make the sky appear bluer. (0 no ozone, 1 clear day atmosphere, 2 city like atmosphere). higher value for bluer sky.
 cloud_density: density of the cloud, varying from 0 to 0.04. (0.01 very thick cloud, 0.04 very heavy cloud)
 output:  texture color output."

Below is an example bout how to make function calls to model the scene to fit the description:
 Question:  given the text description of the scene: "The river, reflecting the clear blue of the sky, glistened like a silver ribbon as it wound its way through the lush valley, its tranquil waters whispering secrets to the ancient trees." analysis the function parameter and call the function to generate the sky.
 Solution: From the description of "clear blue sky", the sun_intensiy can not be low, let's set it as "median";As the sky is blue, the sun_elevation cannot be very low (not morning/sunset/evening), we can set it to 50.
 The sun rotation does not affect the sky appearance, let's set it to 0. To fit the description of "clear blue". The air_density and dust_density should be low, we can set air_density to 1, dust_density to 0.  To make the sky more blue, we can increase the ozone value, let's make it as 2.
 As the sky is very blue and clear, the cloud density should be low, we can set it as 0.0005. We can model the sky by calling the following function:
 ```python sky_texture_node(sun_intensity = 'median',sun_elevation=50,sun_rotation=0,air_density=1,dust_density=0, ozone=2,cloud_density=0.0005)```
.
 Question: Given the text description: "A peaceful clearing in a dense forest, blanketed in fresh, untouched snow. Tall birch trees, their white bark standing out starkly against the dark evergreens and the gray, overcast sky, form a natural cathedral. In this peaceful clearing, the scene is bathed in the soft glow of early morning light. The sun, just beginning to rise above the horizon, casts a gentle, golden hue across the landscape. Its intensity is subdued, as if still gathering strength to break through the thick canopy of trees that surround the clearing. The air in the clearing is crisp and cool, carrying with it a hint of winter's bite. It is still and calm, devoid of any breeze or gusts that might disturb the tranquility of the scene. The air feels pure and invigorating, as if it has been untouched by human presence.
Looking up towards the sky, the overcast conditions create a muted, gray backdrop. The clouds hang low, their edges blending seamlessly with the horizon, creating a sense of enclosure within the clearing. The sky appears vast, yet somehow intimate, as if it is a part of the natural cathedral formed by the towering birch trees.
The cloud density is moderate, with the clouds spread out across the sky in a gentle, billowing fashion. They are not heavy or oppressive, but rather wispy and ethereal, adding a touch of mystery to the scene. As the day progresses, the clouds shift and change shape, creating a dynamic and ever-evolving canvas above the clearing.
Overall, the daytime in this clearing is characterized by a serene ambiance, with the sun's gentle rays filtering through the dense forest and the sky adorned with a delicate veil of clouds. It is a scene that invites contemplation and reflection, a sanctuary of tranquility amidst the wild beauty of nature." analysis the function parameter and call the function to modelling the sky appearance using nishita method.

Based on the description, we can analyze the function parameters as follows:
- sun_intensity: The description mentions a "soft glow of early morning light" and the sun's intensity is described as "subdued". Let's set the sun_intensity to 'low'.
- sun_elevation: The sun is described as "just beginning to rise above the horizon". Let's set the sun_elevation to -5.
- sun_rotation: The description does not provide any specific information about the sun's rotation, so we can set it to 0.
- air_density: The air is described as "crisp and cool" and "untouched by human presence". Let's set the air_density to 0.
- dust_density: The description does not mention any dust or particles in the air, so we can set the dust_density to 0.
- ozone: The description does not provide any specific information about the ozone density, so we can set it to 1.
- cloud_density: The clouds are described as "moderate" and "spread out across the sky in a gentle, billowing fashion". Let's set the cloud_density to 0.02.
We can model the sky by calling the following function:
```python
sky_texture_node(sun_intensity='low', sun_elevation=-5, sun_rotation=0, air_density=0, dust_density=0, ozone=1, cloud_density=0.02)
```

Figure 12: **Communication Between System and Modeling Agent.**

**Document:**
  **Explanation:** The function takes a built natural scene as input and adds base trees to it.
  **Inputs:**
    scene: The built natural scene.
    density(float): The density of the trees.
    distance_min(float): Minimum distance between trees.
    leaf_type(string): The type of leaf on the tree. Select one from the list ['leaf', 'leaf_broadleaf', 'leaf_ginko', 'leaf_maple', 'flower', 'None'].
'Leaf' allows further custom settings like changing the leaf shape and color. 'Leaf_broadleaf', 'leaf_ginko', and 'leaf_maple' build the leaves with
predefined shapes: broad, ginko, and maple respectively. 'Flower' will create flowers on the tree instead of leaves. 'None' will not generate leaves on
the tree.
    fruit_type(string): The type of fruit on the tree. Select one from the list ['apple', 'blackberry', 'coconut_green', 'durian', 'starfruit', 'strawberry',
'custom_fruit', 'None']. Each of the fruit types will create the corresponding fruit on the tree. 'Custom_fruit' can create customizable fruit on the tree
with further adjustments. If the desired fruit is not on the list, 'custom_fruit' should be chosen. 'None' will not generate any fruit on the tree.

**Code:**
```python
import TreeFactory
def add_trees(scene, density, distance_min, leaf_type, fruit_type, leaf_density):
    trees = TreeFactory.create(density, leaf_type, fruit_type, leaf_density)  # create trees
    scene.placement(trees, distance_min)  # place trees in the scene
    return
```

**Information:**
  "Possible trees in the scene, tree types, leaf density, possible seasons, tree size, appearance of the tree branch, branch type, and leaf color."

**Example:**
  **Question:** Given the text description of the scene, analyze the function parameters and call the function to add more trees that fit the text
description.
  **Solution:** From the text description, three different types of trees are identified in the scene: birch, dogwood, and Japanese maple. From "a cluster
of," we infer the tree density is relatively high; let's set it to 0.2. Birch tree leaves are typically oval or triangular, so let's set the leaf type to 'leaf' to
allow further customization. There are no fruits mentioned for the birch tree in the description. Given the scene likely depicts autumn, the leaf_density
should not be too high; let's set it to 0.02. For the dogwood tree, as the leaves are described as oval or elliptical with smooth edges, the leaf type is set
to 'leaf'. It is mentioned that "clusters of small, vibrant berries cling to its branches," so we can choose the fruit_type as 'blackberry'. The
leaf_density is set to 0.02. For the Japanese maple tree, the leaf type is set to 'leaf_maple'. No fruits are mentioned for the maple tree. As the scene
likely takes place in autumn, the leaf density for the maple tree should be high; let's set it to 0.1. The function can be called three times to create three
different trees:

```python
add_trees(scene, density=0.2, distance_min=1, leaf_type='leaf', fruit_type='None', leaf_density=0.02)  # create birch trees
add_trees(scene, density=0.1, distance_min=1, leaf_type='leaf', fruit_type='blackberry', leaf_density=0.02)  # create dogwood trees
add_trees(scene, density=0.1, distance_min=1, leaf_type='leaf_maple', fruit_type='None', leaf_density=0.1)  # create Japanese maple trees
```

Figure 13: **Prompt Example of Adding Trees.**

Document:
**Input:**
   sun_intensity: Multiplier for sun disc lighting. (Choose from 'low', 'median', 'high')
   sun_elevation: Rotation of the sun from the horizon (in degrees). (0: sunset, sun rising, 90: daytime, -10: night)
   sun_rotation: Rotation of the sun around the zenith (in degrees).
   air_density: Density of air molecules. (0: no air, 1: clear day atmosphere, 2: highly polluted day)
   dust_density: Density of dust and water droplets. (0: no dust, 1: clear day atmosphere, 5: city-like atmosphere, 10: hazy day)
   ozone: Density of ozone molecules; useful to make the sky appear bluer. (0: no ozone, 1: clear day atmosphere, 2: city-like atmosphere). A higher value yields a bluer sky.
   cloud_density: Density of the clouds, ranging from 0 to 0.04. (0.01: very thick cloud, 0.04: very heavy cloud)
**Output:**
   texture_color_output: The output texture color of the sky.

Code:
```python
import nishita_sky_modelling
def sky_texture_node(sun_intensity, sun_elevation, sun_rotation, air_density, dust_density, ozone, cloud_density):
    texture = nishita_sky_modelling.model_sky(sun_intensity, sun_elevation, sun_rotation, air_density, dust_density, ozone, cloud_density)
    return texture
```

Information:
   The time of day depicted in the scene (morning/noon/evening), the possible season, the condition of the air, the blueness of the sky, and the cloud density.

Example:
**Question:**
   Given the text description of the scene: "The river, reflecting the clear blue of the sky, glistened like a silver ribbon as it wound its way through the lush valley, its tranquil waters whispering secrets to the ancient trees," analyze the function parameters and call the function to generate the sky.

**Solution:**
   From the description of a "clear blue sky," we infer that sun_intensity cannot be low, let's set it as "median". As the sky is clear and blue, the sun_elevation must not be very low, indicating it's not morning or evening; we can set it to 50 degrees.
   Sun rotation does not impact the appearance of the sky significantly; thus, we will set it to 0 degrees. To align with the description of "clear blue," we should choose lower values for air_density and dust_density. Hence, we will set air_density to 1 and dust_density to 0. To enhance the blueness of the sky, we can increase the ozone value to 2.
   As the description suggests a clear, blue sky, the cloud density must be low. Therefore, we will set it to 0.0005. The sky can be modeled by invoking the function as follows:

```python
sky_texture_node(sun_intensity='median', sun_elevation=50, sun_rotation=0, air_density=1, dust_density=0, ozone=2, cloud_density=0.0005)
```

Figure 14: **Prompt Example of Sky Modeling Function.**

(a). *"The sun shines brightly in a clear blue sky. Not a cloud in sight."*

(b). *"Thick gray clouds cover the entire sky, diffusing the sunlight."*

(c). *"The sky appears hazy, with a soft, diffuse sunlight filtering through."*

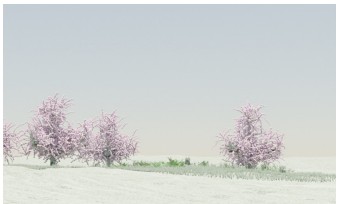 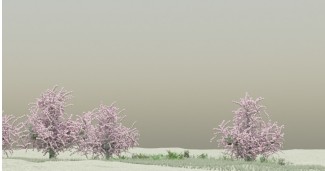 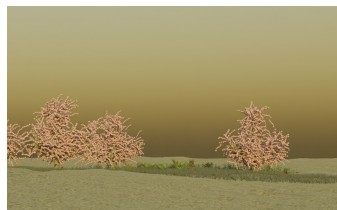

(d). *"The sky is painted in shades of orange and pink as the sun sets behind the island."*

(e). *"The sky is a mixture of soft purples and pinks as the sun rises."*

(f). *"A canopy of stars fills the dark night sky, with the island bathed in moonlight."*

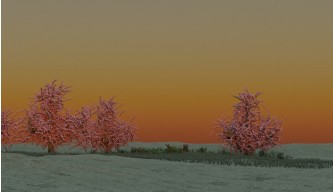 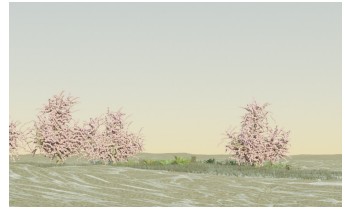 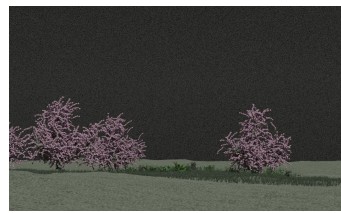

(a). Solution: From the description of "clear blue sky" and "not a cloud in sight", we can infer that the sun_intensity should be high, the sun_elevation should be around 90 (daytime), the sun_rotation can be set to 0. The air_density and dust_density should be low, so we can set them to 1 and 0 respectively. Since there are no clouds, the cloud_density should be 0. To make the sky appear bluer, we can increase the ozone value, so let's set it to 2. We can model the sky by calling the following function:
```python
sky_texture_node(sun_intensity='high', sun_elevation=90, sun_rotation=0, air_density=1, dust_density=0, ozone=2, cloud_density=0)
```

(b). Solution: From the description of "Thick gray clouds cover the entire sky", the sun_intensity should be low to account for the diffused sunlight. Let's set it as "low". The sun_elevation should be high to indicate that it is daytime. Let's set it to 90. The sun_rotation does not affect the sky appearance, so we can set it to 0. The air_density and dust_density should be high to represent the presence of thick clouds. Let's set air_density to 2 and dust_density to 5. Since the sky is covered with thick gray clouds, the ozone value can be low. Let's set it to 0. As mentioned, the entire sky is covered with thick gray clouds, so the cloud_density should be high. Let's set it to 0.04. We can model the sky by calling the following function:
```python
sky_texture_node(sun_intensity='low', sun_elevation=90, sun_rotation=0, air_density=2, dust_density=5, ozone=0, cloud_density=0.04)
```

(c). Solution: From the description of "hazy sky" and "soft, diffuse sunlight", we can infer that the sun_intensity should be low. Let's set it as "low". The sun_elevation should be low as well, indicating a time close to sunrise or sunset. Let's set it to -10. The sun_rotation does not affect the sky appearance, so we can set it to 0. To fit the description of a hazy sky, the air_density and dust_density should be higher. Let's set air_density to 2 and dust_density to 5. The ozone value can be set to 1 to maintain a natural sky color. Since the sky is hazy, the cloud_density should be higher as well. Let's set it to 0.02 to represent a moderately cloudy sky. We can model the sky by calling the following function:
```python
sky_texture_node(sun_intensity='low', sun_elevation=10, sun_rotation=0, air_density=2, dust_density=5, ozone=1, cloud_density=0.02)
```

(d). Solution: From the description of "shades of orange and pink", the sun_intensity can be set as "low" to create a softer and warmer lighting. As the sun is setting, the sun_elevation can be set to 0. The sun_rotation does not affect the sky appearance, so it can be set to 0. To fit the description of "shades of orange and pink", the air_density and dust_density should be low, so we can set air_density to 1 and dust_density to 0. To create a warmer and more vibrant sky, we can set the ozone value to 1. As the sun is setting, there might be some clouds in the sky, so we can set the cloud_density to 0.01. We can model the sky by calling the following function:
```python
sky_texture_node(sun_intensity='low', sun_elevation=0, sun_rotation=0, air_density=1, dust_density=0, ozone=1, cloud_density=0.01)
```

(e). Solution: From the description of "soft purples and pinks", the sun_intensity can be set to "low" to create a softer lighting effect. Since the sun is rising, the sun_elevation can be set to a low value, such as 10. The sun_rotation does not affect the sky appearance, so it can be set to 0. To create a purple and pink sky, the air_density and dust_density can be set to low values, such as 1 and 0. The ozone value can be set to 0 to avoid making the sky bluer. Since the sky is described as a mixture of colors, the cloud_density can be set to a low value, such as 0.001. We can model the sky by calling the following function:
```python
sky_texture_node(sun_intensity='low', sun_elevation=10, sun_rotation=0, air_density=1, dust_density=0, ozone=0, cloud_density=0.001)
```

(f). Solution: From the description of "dark night sky" and "canopy of stars", we can assume that the sun is not visible in the sky. Therefore, the sun_intensity can be set to 'low'. Since it is a night scene, the sun_elevation can be set to -10. The sun_rotation does not affect the sky appearance, so it can be set to 0. As it is a dark night sky, the air_density and dust_density can be set to 0. The ozone value can also be set to 0 as there is no mention of blueness in the description. Since there are no clouds mentioned, the cloud_density can be set to 0. We can model the sky by calling the following function:
```python
sky_texture_node(sun_intensity='low', sun_elevation=-10, sun_rotation=0, air_density=0, dust_density=0, ozone=0, cloud_density=0)
```

Figure 15: **Single Function Control Result.** Visual result (top) and modeling agent response example (bottom). Our method demonstrates a high degree of accuracy in inferring algorithm parameters, even when they do not possess a direct connection to visual appearance.

