# OpenReview forum: "3D-GPT: Procedural 3D Modeling with Large Language Models"
_ICLR.cc/2024/Conference — Submitted to ICLR 2024_

### Official Review · Reviewer_967x · 2023-10-29

**Soundness:** 3 good
**Presentation:** 3 good
**Contribution:** 1 poor
**Rating:** 3
**Confidence:** 4

**Summary:**

This paper presents a new 3D scene generation pipeline using procedural 3D modeling together with LLMs. Specifically, LLMs are given the documentations of a procedural 3D modeling tools together with human instructions. Three LLM agents, namely the task dispatch agent, the conceptualization agent and the modeling agent, are designed to work together and generate Python scripts for the 3D modeling tools that generate 3D scenes/ objects that corresponds to the given instructions. Qualitative and quantitative experiments are performed to show the results of such pipeline and prove the effectiveness of using three agents to collaborate on this task.

**Strengths:**

1. This paper shows the potential of using LLMs and procedural 3D modeling tools on text-guided 3D generation tasks.
2. Having three agents collaborating on generating the final Python scripts is interesting and proven to be effective.

**Weaknesses:**

1. Though it is a nice application to use LLM with procedural 3D modeling tools for text-guided 3D generation, I think the overall contribution of this paper is not enough to be considered for publication on ICLR. The community has had many similar discoveries on the ChatGPT/ GPT-4's ability on generating parameters for images [1] or 3D objects [2] in a zero-shot or in-context learning way. Therefore, there is no surprise that combining LLM with a better parametric 3D modeling tool, e.g., InfiniGen, could produce better visual results. I think all the three potential directions mentioned in the last section are promising and valuable challenges to solve, which would bring more contribution to the community.

2. The evaluation is limited. I must admit that for a new task of text-guided 3D scene generation, it is hard to construct baseline methods. But at least, there are similar efforts in the community of text-guided 3D object generation that worth being compared with. For example, score-distillation-based methods, e.g., DreamFusion, can be used as baselines for the "Single Class Control" experiments.

[1] Bubeck, Sébastien, et al. "Sparks of artificial general intelligence: Early experiments with gpt-4." arXiv preprint arXiv:2303.12712 (2023).
[2] https://twitter.com/gd3kr/status/1638149299925307392

**Questions:**

See above weaknesses.

---

> ### Author Response · Authors · 2023-11-22
>
> 1. Similar discoveries on ChatGPT/ GPT-4's ability on generating parameters for images [1] or 3D objects [2].  No surprise that combining LLM with a better parametric 3D modeling tool InfiniGen could produce better visual results.
>
> We appreciate the reviewer's comments regarding the perceived limitations of our contribution. However, we respectfully disagree with the assertion that our work lacks novelty. It is essential to differentiate between image generation [1] and 3D modeling, as these areas pose distinct challenges.
>
> Moreover, the 3D examples provided by [2] do not reflect the complexities we address. Such a way to generate Blender Python code lacks a comprehensive understanding of API functions and their alignment with natural language. The generated Python codes would have high error rates. Yet, it lacks mechanisms for verifying code executability and correctness. Furthermore, directly generating Python codes does not effectively manage dependencies between 3D contents. For instance, when generating trees, the spatial relationship between the leaf and tree branches is not explicitly modeled. As a result, this method of applying LLMs to Blender code could produce inadequate 3D models, especially in the context of complex large-scale scenes.
>
> Also, our research demonstrates that a simple combination of LLMs with the procedural generation library InfiniGen leads to subpar results.
> Note that, while the idea of combining LLMs with procedural 3D modeling may now appear straightforward in hindsight, creating an effective workflow is far from simple. Our solution involves three distinct agents: the Task Dispatch Agent interprets and chooses the right API calls, the Conceptualization Agent enhances the scene description with adequate details, and the Modeling Agent comprehends how textual descriptions translate into specific API parameters. This multi-agent approach ensures effective and detailed 3D model generation from textual inputs. Please refer to the results presented in Section 4.2 (Table 1, Figure 6) of our paper, as well as the **updated** supplementary materials (Section 6.2, Table 2, Table 3). These results clearly illustrate the significant roles played by each component within our framework in enhancing the final output.
>
> 2. Comparison with Previous Text-to-3D Methods:
>
> In response to the reviewer's request for a comparison with previous text-to-3D methods, we have included an additional comparison with Dreamfusion [a]  in the **supplementary materials (Section 6.3, Figure 7)**. We show that while Dreamfusion generates reasonable 3D for single objects, it struggles to generate large-scale natural scenes, as demonstrated in our **supplementary materials (Section 6.3, Figure 8)**. In comparison, our method is effective in both scenarios, showing the advantages of our approach in handling more complex 3D modeling tasks. Furthermore,  our framework supports sequential scene editing without necessitating re-training.
>
> Furthermore, it's worth noting that previous text-to-3D methods encountered challenges when generating large-scale natural scenes, as demonstrated in our **supplementary materials (Section 6.3, Figure 8)**.  These observations emphasize the strengths and advantages of our approach in handling more complex 3D modeling tasks.
>
> [a] DreamFusion: Text-to-3D using 2D Diffusion. In ICLR 2023
>
> Thank you for your valuable feedback and insightful comments. We truly appreciate the time and effort you dedicated to reviewing our work. Please let us know if any questions.

---

### Official Review · Reviewer_m12k · 2023-11-01

**Soundness:** 3 good
**Presentation:** 4 excellent
**Contribution:** 2 fair
**Rating:** 3
**Confidence:** 4

**Summary:**

The authors present a method that can take in natural language and output code in a Domain-Specific Language. The DSL is a system that can generate 3D scenes and assets in a procedural fashion.

Therefore the overall pipeline converts natural language into realistic, high quality 3D scenes.

**Strengths:**

The results are **incredibly strong** - looking at the submitted video. Especially the results on sky-editing.

**Weaknesses:**

**(MAJOR) Lack of detail**

---

The paper describes the system at a very high level. We are introduced to the "Task Dispatch Agent", the "Conceptualization Agent" and the "Conceptualization Agent" but no details of how they are actually implemented. There is no detail of what subset of the InfiGen language is used, no detail of “translate the scene into a winter setting”, it pinpoints functions like add snow layer() and update trees()"

While there are some examples in the Appendix of the prompts and the associated code, it is woefully incomplete.

In the current iteration, it is almost impossible to implement/reproduce the paper.

I will not knock the paper down for a lack of quantitative results because this space is very new and no metrics apart from user studies really exist and creating a new procedural baseline would itself be a lot of work.

**Questions:**

I wonder if the authors would send this work to a compilers conference/journal instead?

In case they are too rigorous to accept LLM based papers, why not something like ToG or SIGGRAPH. Both have wonderful papers describing procedural systems where the parameters come from some heuristic model. This paper with its strong results and a graphics focus seems like a perfect fit for such venues.

I would be willing to accept this paper only after a major rewrite which would include a lot more description of the various components in the proposed approach.

---

> ### Author Response · Authors · 2023-11-23
>
> 1. Submission to other venues (e.g., SIGGRAPH)
> We appreciate your suggestion to consider SIG and TOG for our paper submission. While we agree that our work aligns with the themes of these conferences, we strongly believe that our paper is also highly relevant for ICLR, fitting squarely within its broad scope of machine learning topics. Our work uniquely intersects with the domain of large language models (LLMs) and AI agents, areas of keen interest within the ICLR community.
>
> The core of our contribution lies in a novel approach to formulating the text-to-3D problem. We introduce 3D-GPT to overcome significant hurdles in 3D scene creation. This includes generating expansive scenes, animating 3D models, maintaining coherence in 3D spaces, and creating intricate details in 3D objects.These are issues that previous methodologies have struggled with and are of considerable interest to ICLR readers and authors.
>
>
> Furthermore, our future directions, including automatically generating procedural rules and fine-tuning models, align well with the interests of the ICLR community and require machine learning-based solutions that this community can provide.
>
> 2. Describe the system in a very high-level, impossible to implement the paper.
>
> We have significantly revised Section 3 of the paper to provide more details on the various components of our method in a concise, yet reproducible way. We highlight the key changes as red color. Extended full examples of communication between agent, scene generation script and example implementation for modeling agent are included in the supplementary material. As mentioned elsewhere we will release all code and examples on acceptance. **The revised version of the paper has been uploaded. : )**
>
> We want to thank you for your helpful comments and suggestions, which really improved our paper. We're grateful for your patience and commitment during the review process.

---

### Official Review · Reviewer_nMkH · 2023-11-02

**Soundness:** 3 good
**Presentation:** 3 good
**Contribution:** 3 good
**Rating:** 6
**Confidence:** 3

**Summary:**

This paper Introduces 3D-GPT, a training-for-free framework designed for 3D scene generation, which generates Python codes to control 3D software, potentially offering increased flexibility for real-world applications.

**Strengths:**

1. The generation ability of 3D scenes and objects are good according to the demo.

2. The step-by-step refinement of 3D outputs is meaningful and impressive.

3. The paper is easy to follow with good-quality figures.

**Weaknesses:**

I'm not an expert in 3D procedural 3D generation, and here are some of my concerns.

1. The paper only shows examples of 3D plants and forests. Can 3D-GPT work on other objects or scenes, such as human and street? Can 3D-GPT generalize well to more complex scenarios?

2. The paper adopts ChatGPT as LLM. How about other open-source LLMs, such as Alpaca, LLaMA-Adapter, or Vicuna?

**Questions:**

See weakness

---

> ### Author Response · Authors · 2023-11-22
>
> 1. Expanding the Scope of 3D-GPT Beyond Plants and Forests:
>
> Our method, which leverages the procedural algorithm library Infinigen, was originally designed for natural scene generation. However, its versatility extends beyond this specific application. Its key strength lies in its ability to proficiently handle various 3D modeling tasks in diverse contexts without necessitating fine-tuning. Our system operates in a zero-shot manner, independently processing each modeling function by the conceptualization and modeling agents. This independence ensures that our method can readily adapt to new functions, provided the requisite information. We will provide more complex object modeling in the supplementary before the rebuttal deadline.
>
> 2. Evaluation with Alternative Open-Source LLMs:
>
> We have also conducted evaluations of our approach using Llama2/GPT4, as outlined in the **supplementary materials (Section 6.2, Table 3)**. Our results demonstrate that the performance of our method remains consistent when applied across different LLMs, underscoring its robustness and broad applicability.
>
> Thank you for your valuable review. Your feedback is much appreciated, and we're committed to addressing your concerns. We hope our explanations have addressed your concerns and we remain open to further discussion.

---

### Official Review · Reviewer_QYLZ · 2023-11-10

**Soundness:** 3 good
**Presentation:** 3 good
**Contribution:** 2 fair
**Rating:** 5
**Confidence:** 4

**Summary:**

This paper proposes a workflow for procedural 3D scene generation conditioned on text descriptions using pre-trained LLMs. It leverages an existing procedural generator, InfiniGen, to create 3D contents, and uses LLMs to pick a set of procedural functions from InfiniGen and infer their corresponding parameters given the text description of the scene. The authors split the task into three steps (agents): the first step is to select the set of functions given the prompt, the second step is to infer more detailed descriptions given the required informatin, and the last step is to generate the parameters for each function given the detailed description. The experiment results show that this workflow can produce single class objects with details and complex scenes.

**Strengths:**

- The proposed method does not require training.
- The multi-agent approach is effective. It might share similar advantages as other methods such as chain-of-thought or tree-of-thought, where the final output from the LLM is guided by a curated step-by-step instruction.
- It addresses another potential direction for text-to-3D generation: utilizing tools or existing 3D procedural models.
- It demonstrates the potential of using LLMs to control tools for content creation and 2D/3D modeling.

**Weaknesses:**

- Lack of details about the model and experiment setting and how the results are affected. For example, which LLM is used? What is the size of the function set F? Does the size of F affect the quality? How many examples are provided? Do the example related to the prompt L? Does zero-shot/few-shot make any difference?
- Evaluation can be improved. It would be great to do ablation studies on D, C, I, E, and answer the questions mentioned above.
- It would be great to further explore the limitations and failure cases. For example, does the complexity of the scene affect the results? Does the number of parameters or the design space (e.g., parameter ranges) affect the results?
- The proposed method demonstrates that the LLM can convert the input prompts to python codes that controls the functions and parameters. However, since the 3D modeling capability comes from the procedural models, not from the LLM, it seems the task is closer to scene composition or object inference instead of modeling. It will be more interesting to see if LLMs can generate 3D modeling commands or procedural modeling sequences/rules.

**Questions:**

- There are many procedural models available for Blender. It will be interesting to see if this workflow will work on any arbitrary procedural models given the same amount of information, i.e., D, C, I , and E, such as house, car, airplanes instead of focusing on scenes in InfiniGen.

- It will be great to see if the selected functions can formulate some dependencies, for example, to generate 'flowers on the trees', the trees need to be created first and the positions of the flowers are based on the positions of the tree branches.

- It will be great to see a full example of the input and output of each agent in the process.

- Have the authors tried to also provide the functions picked by the TDA to CA?

- It is obvious that this method will help users who know nothing about 3D modeling, but I am curious whether the sequential editing task in Fig 4 will be more efficient to professional Blender users (e.g., game developers) to achieve a desired outcome, compared to tweaking the parameters by themselves.

---

> ### Author Response · Authors · 2023-11-22
> **Lack of Detail**
>
> **Lack of Detail:**
> 1. Which Large Language Model (LLM) is used?
>
> We utilized GPT-3.5 for all experiments.
>
> 2. What is the size of the function set F?
>
> The function set F comprises all functions and subfunctions detailed in this script (https://github.com/princeton-vl/infinigen/blob/main/worldgen/generate.py), except for the 'creatures' class. These functions are integral to our scene creation process.
>
> 3. Does the size of F affect the quality of generated scenes?
>
> Yes, the size of F directly impacts scene quality. Infinigen's scene generation employs a fixed function set. Removing a function, such as 'flower_generation,' eliminates the ability to create corresponding elements (e.g., flowers) in the scene. Similarly, omitting a sub-function like 'control_flower_petal' restricts control over specific aspects (e.g., flower petal movement), impacting scene controllability and diversity.
>
> 4. How many examples are provided for each function?
>
> We provide one example per function.
>
> 5. Is the provided example related to the user's prompt L?
>
> No, the prompt L is user-supplied. Our examples, one per function, might occasionally resemble some user inputs L, but they are not directly related.

---

> ### Author Response · Authors · 2023-11-22
> **Evaluation Improvement**
>
> **Evaluation Improvement**
>
> 1. Ablation Studies on D, C, I, E: Impact of Zero-shot/Few-shot Learning.
>
> Thank you for your insightful comments. In response, we have updated our submission with comprehensive ablation studies in the supplementary material (Section 6.2) . Table 1 elucidates the roles of D, C, I, and E components. Absence of function documentation (D) impairs our method's ability to interpret function input parameters, thereby reducing the CLIP Score and elevating the Failure Rate. Without code (C), while the system grasps basic function and parameter knowledge, it experiences a decline in CLIP Score and an increased Failure Rate. The lack of information (I) hinders the conceptualization agent's capacity to enrich text with vital details, leading to a significant drop in CLIP Score. Excluding examples (E), as shown in Table 4, markedly diminishes the CLIP Score and heightens the Failure Rate. We noted no notable difference in performance between one-shot, two-shot, and three-shot prompts, though introducing more examples tends to reduce parameter diversity.
>
> 2. Impact of Scene Complexity and Parameter Count on Results.
>
> The outcomes are indeed influenced by scene complexity and parameter count. The definition of scene complexity matters: if the initial text input features complex objects beyond our procedural algorithm's scope, the quality of generated content is limited. Conversely, detailed initial text descriptions enhance the conceptualization agent's understanding, yielding results more aligned with the input. The number of parameters is critical; removing functions or sub-functions (e.g., 'flower_generation' or 'control_flower_petal') limits modeling abilities, affecting controllability and diversity. Assessing the impact of function set size and parameter count is complex due to varying function significance. For example, removing 'control_flower_petal()' may not markedly affect the CLIP score, but eliminating 'terrain_generation()' substantially reduces it, as terrain modeling becomes impossible. Consequently, we opted not to include evaluations of this aspect, but we will update some visual example in supplementary before rebuttal deadline.

---

> ### Author Response · Authors · 2023-11-22
> **Others: Suggestions and Queries**
>
> **Others: Suggestions and Queries**
>
> 1. LLMs in 3D Modeling and Procedural Sequences:
>
> We value your suggestion about leveraging LLMs for procedural rules in 3D modeling, a topic we've earmarked for future exploration in our paper. Our current research primarily focuses on utilizing LLMs for efficient function control and parameter inference in 3D modeling. This approach is pivotal to our overarching aim of reducing human intervention in the 3D modeling process through advanced LLM applications.
>
> 2. Function Dependencies and Complex Tasks:
>
> The functions we have selected do establish dependencies. For example, the generate_trees() function can autonomously design tree branches, calculate surfaces programmatically, and accurately place leaves and flowers on these surfaces. This ensures correct dependencies and spares LLMs from the intricate task of determining precise 3D leaf/flower locations, a challenging feat for pretrained LLMs.
>
> 3. Availability of Full Examples of Input and Output:
>
> We recognize the value of detailed examples for each agent, yet their inclusion would overly extend our manuscript. To keep our narrative concise, we will showcase these examples through supplementary videos or within our user interface. This approach ensures clarity and accessibility without overburdening the text. We will release the code that contains all examples.
>
> 4. Adaptability to Arbitrary Procedural Models:
>
> Our system functions in a zero-shot manner, with each function processed independently by the conceptualization and modeling agents. This independence ensures that our method can adapt to new functions as long as the necessary information is supplied.
> As shown in our paper PAGE 5 bottom block, with provided code C, example E, and function document D, our method can understand the function and infer the required parameters.
>
> 5. Integration of Functions Selected by TDA in CA:
>
> To be clear, in our current pipeline, the conceptualization agent processes only those functions that are selected by the task dispatch agent.
>
> 6. Efficiency for Professional Blender Users:
>
> For Blender experts that not familiar with procedural generation, in the scenario depicted in Figure 4, transforming flat terrain into mountainous landscapes would require manually rebuilding terrain and repositioning each element, which is time-intensive.
> For Blender experts that are familiar with infinigen libraries might find this task straightforward. However, for each new procedural library, the experts are required to understand every new function, it also poses a significant challenge for professional Blender users to quickly grasp and utilize every new procedural library and its functions. Our approach aims to simplify and expedite this process.

---

> ### Author Response · Authors · 2023-11-22
>
> Thank you sincerely for sharing your invaluable review with us, and for your insightful feedback on how we can improve our evaluations. Your input is highly valued, and we are fully dedicated to addressing your concerns to the best of our abilities. Please feel free to reach out if you have any additional questions or if there is anything else we should provide.

---

> > ### Comment · Reviewer_m12k · 2023-11-22
> > **Confirmation of review**
> >
> > I thank the authors for the rebuttal and the updated manuscript.
> >
> > I really liked the paper but even the revised manuscript is almost impossible to reproduce.
> >
> > In addition, I would seriously like to know how the paper is any different than [1] but extended to a 3d setting (and procedural models).
> >
> > I retain my score.
> >
> > [1] https://viper.cs.columbia.edu/

---

> > > ### Author Response · Authors · 2023-11-23
> > > **Thank you for your new comments**
> > >
> > > Dear Reviewer m12k,
> > >
> > > Thank you for your insightful reviews and new comments on our paper. We appreciate your time and dedication to the review process. Very sorry for the delay in responding to your feedback. We are currently in the process of diligently addressing all of your comments as well as undertaking a comprehensive rewrite of the entire paper. Both the response and the updated paper will be uploaded within the next few hours.
> > >
> > > Thank you again for your valuable feedback and your patience throughout this process. Please stay tuned!

---

> > ### Comment · Reviewer_QYLZ · 2023-11-23
> >
> > Thank you for the detailed rebuttal and updated manuscript. However, I still think this paper is focused on using LLMs for controlling and leveraging existing procedural models for scene and object compositions instead of the procedural 3D modeling capability. I will remain my score as 5.

---

> ### Author Response · Authors · 2023-11-23
> **TO REVIEWER M12K, Difference between our work and ViperGPT**
>
> We thank you for bringing our attention to the contemporary work of ViperGPT. We would like to underscore the distinctive features that set our work apart from theirs:
>
> **Task Complexity:** Our approach tackles the more intricate challenge of 3D modeling. To address this complexity, we have implemented multiple agents that collaboratively handle various subtasks, resulting in a cooperative solution. This stands in contrast to ViperGPT, which lacks the incorporation of agents and cooperative strategies.
> In particular, ViperGPT only provides its agent with function names. In our case, we have a more extensive library with intricate functions, such as the sky modeling function we showcased in the paper Figure 5. This function demands a deep understanding of the Nishta sky modeling algorithm for parameter inference. If we were to apply their approach to our work, it would be inadequate for a pre-trained language model to comprehend these functions fully and, consequently, to make informed selections regarding the relevant functions and arguments based on the given instructions.
>
> **Scope of Focus:** While both our work and ViperGPT share a common vision of employing a LLM to generate executable Python code, our emphasis diverges in the specific goal and methodology. Our work is primarily centered around 3D generation, whereas ViperGPT is oriented towards visual understanding. Moreover, our work employs multiple agents to solve the task.
> In light of these distinctions, we believe our work makes a unique contribution to the field, providing advancements in handling complex 3D modeling tasks through collaborative agent-based strategies. We will reference ViperGPT in the related work of our paper and include a description of the differences.
>
> Thank you once again for your valuable feedback. We are dedicated to addressing any remaining concerns and ensuring the clarity and significance of our contributions.

---

> > ### Comment · Reviewer_m12k · 2023-11-23
> > **Response**
> >
> > Thank you for your response.
> >
> > I agree that ViperGPT is focussed in image analysis instead of generation.
> >
> > However, the underlying ideas are very similar - given a pretrained LLM, design a good prompt so that the LLM can convert natural language to a DSL - InfiniGen in your case, something else in the case of ViperGPT.
> >
> > > In particular, ViperGPT only provides its agent with function names.
> >
> > This is incorrect in my view, as the prompt to ViperGPT is a prompt consisting of the function call, it’s documentation and a few examples so that the LLM “understands” the function and it’s parameters.
> >
> >
> > > To address this complexity, we have implemented multiple agents that collaboratively handle various subtasks, resulting in a cooperative solution
> >
> > I like the paper, however the description of these components is at a very high level meaning I cannot exactly discern how each of the components contributes to the overall solution. Or how the work is different from ViperGPT.
> >
> > The paper to me looks to have very solid contribution with very solid results. It is just that in the current state of writing, it is difficult to know exactly what was implemented and how it was implemented.
> >
> > It might be too late but you could easily give your gpt model a prompt of the infinigen api and look at the results like ViperGPT. It would really help the paper.
> >
> > Looking at the new pseudo code and the new results, I will update my score, if during the discussion phase, other reviewers also agree that the paper is substantially different than ViperGPT.

---

### Public Comment · ~Dazhou_Yu1 · 2025-03-03
**Curious About Subsequence Instruction Editing and Object Understanding in LLM-Driven Environments**

I really appreciate the author’s insightful and impressive work, especially the exploration of Subsequence Instruction Editing. I have a couple of questions regarding Figure 4 and the model’s understanding of generated objects:
1. Why does the background terrain change in Figure 4? Is this a requirement of the subsequent instructions, or is there another reason for this transformation?
2. To what extent does the LLM retain an understanding of previously generated objects? For example, if a tree is created at specific coordinates (x, y, z), does the model recognize that object and allow targeted modifications to it? Or does each editing step simply overwrite or replace existing objects without maintaining a persistent object-level understanding?

Thank you for this great contribution — I’m eager to learn more!

---

### Meta-Review · Area_Chair_7ZVj · 2023-12-10

**Metareview:**

The submission explores the use of large language models for procedural scene generation.  Reviewers liked the idea but were concerned about the limited evaluation and lack of implementation details.  Eventually, three of the four reviewers recommended rejection, and the only marginally positive reviewer acknowledged that they are not an expert in the area.  The AC agreed with the reviewers on the submission's strengths and weaknesses.  While the AC cannot recommends acceptance at this time, the authors are strongly encouraged to revise the submission based on the comments for the next venue.

Note: there was a concern regarding the fit with ICLR, which was dismissed and not considered during the decision-making process.

**Justification For Why Not Higher Score:**

There were shared concerns about the limited evaluation and lack of implementation details.

**Justification For Why Not Lower Score:**

N/A

---

### Decision · Program_Chairs · 2024-01-16

Reject